# Protective Effect of Polysaccharides Extracted from *Cudrania tricuspidata* Fruit against Cisplatin-Induced Cytotoxicity in Macrophages and a Mouse Model

**DOI:** 10.3390/ijms22147512

**Published:** 2021-07-13

**Authors:** Eui-Baek Byun, Ha-Yeon Song, Woo Sik Kim, Jeong Moo Han, Ho Seong Seo, Sang-Hyun Park, Kwangwook Kim, Eui-Hong Byun

**Affiliations:** 1Advanced Radiation Technology Institute, Korea Atomic Energy Research Institute, Jeongeup 56212, Korea; ebbyun80@kaeri.re.kr (E.-B.B.); songhy@kaeri.re.kr (H.-Y.S.); jmhahn@kaeri.re.kr (J.M.H.); hoseongseo@kaeri.re.kr (H.S.S.); 2Functional Biomaterial Research Center, Korea Research Institute of Bioscience and Biotechnology, Jeongeup 56212, Korea; kws6144@kribb.re.kr; 3Department of Food Science and Technology, Kongju National University, Yesan 32439, Korea; shpark@kongju.ac.kr (S.-H.P.); nxkkwxm@nate.com (K.K.)

**Keywords:** *Cudrania tricuspidata*, polysaccharides, cytoprotective action, cisplatin, macrophage, cancer cisplatin

## Abstract

Although cisplatin is one of most effective chemotherapeutic drugs that is widely used to treat various types of cancer, it can cause undesirable damage in immune cells and normal tissue because of its strong cytotoxicity and non-selectivity. This study was conducted to investigate the cytoprotective effects of *Cudrania tricuspidata* fruit-derived polysaccharides (CTPS) against cisplatin-induced cytotoxicity in macrophages, lung cancer cell lines, and a mouse model, and to explore the possibility of application of CTPS as a supplement for anticancer therapy. Both cisplatin alone and cisplatin with CTPS induced a significant cytotoxicity in A549 and H460 lung cancer cells, whereas cytotoxicity was suppressed by CTPS in cisplatin-treated RAW264.7 cells. CTPS significantly attenuated the apoptotic and necrotic population, as well as cell penetration in cisplatin-treated RAW264.7 cells, which ultimately inhibited the upregulation of Bcl-2-associated X protein (Bax), cytosolic cytochrome c, poly (adenosine diphosphateribose) polymerase (PARP) cleavage, and caspases-3, -8, and -9, and the downregulation of B cell lymphoma-2 (Bcl-2). The CTPS-induced cytoprotective action was mediated with a reduction in reactive oxygen species production and mitochondrial transmembrane potential loss in cisplatin-treated RAW264.7 cells. In agreement with the results obtained above, CTPS induced the attenuation of cell damage in cisplatin-treated bone marrow-derived macrophages (primary cells). In in vivo studies, CTPS significantly inhibited metastatic colonies and bodyweight loss as well as immunotoxicity in splenic T cells compared to the cisplatin-treated group in lung metastasis-induced mice. Furthermore, CTPS decreased the level of CRE and BUN in serum. In summation, these results suggest that CTPS-induced cytoprotective action may play a role in alleviating the side effects induced by chemotherapeutic drugs.

## 1. Introduction

Cancer remains a major cause of human death worldwide, and chemotherapy is the main treatment modality for cancer [1]. Epidemiological studies have suggested that cancer remains one of the leading causes of mortality and morbidity despite decades of clinical and basic research on the treatment and prevention of cancer [2,3,4]. Numerous cytotoxic agents including paclitaxel, 5-fluorouracil, doxorubicin, and cisplatin (cisp) have been used in standard chemotherapeutic treatments for improving survival of patients with cancer [5,6]. However, patients with cancer can suffer potential side effects (oral mucositis, hepatotoxicity, neurotoxicity, and cardiotoxicity) due to cancer chemotherapy [7,8]. In particular, cisp can cause the death of immune cells, immune dysfunction, and damage of normal tissue because of its strong toxicity and non-selectivity for cancer cells [9,10]. Furthermore, immunosuppression is often observed in cisp-treated patients, which can leads to a state susceptible to infection [11]. Therefore, the protection of immune cells is an important consideration in chemotherapy [12].

Recently, the use of various natural products as well as cancer chemo-preventive agents has attracted widespread attention for cancer therapeutics in preclinical and clinical studies [13,14]. Several studies have reported the role of natural products in regulating chemotherapy-induced cytotoxicity and side effects that may be intolerable to patients with cancer [15]. Several preclinical studies (cell and animal) and clinical trials have revealed that natural products such as curcumin and those obtained from ginseng, ginger, and aloe can potentially reduce chemotherapy-induced cytotoxicity and side effects [16]. Among the biological components derived from various plants, polysaccharides have attracted attention for the alleviation of cytotoxicity induced by toxic substances, as well as for immunomodulation via balanced T helper type 1 cell (Th1) and Th2 potentiating activity [17,18,19]. *Cudrania tricuspidata* is a plant belonging to the family *Moraceae* distributed in East Asia; it has various medicinal and nutritional properties such as anti-inflammatory, neuroprotective, immunomodulatory, and antioxidant effects [20,21,22]. A water-soluble polysaccharide extracted from *C. tricuspidata* root showed cytoprotective effects against immune cell death induced by staurosporine, an initiator of apoptosis, as well as the immunomodulatory activities of macrophages [20]. The cytoprotective effect exerted by natural products was mainly mediated by inhibiting apoptotic signaling pathways induced by chemotherapeutic agents such as cisp, one of the most widely used anticancer drugs [14,23]. Several studies have reported the cytoprotective effects of polysaccharides extracted from *C. tricuspidata* using various toxic models; however, there is no evidence supporting the cytoprotective role of *C. tricuspidata*-derived polysaccharides (CTPS) against cisp-induced cytotoxicity in normal cells and tissues.

Therefore, this study aimed to evaluate the cytoprotective effects of CTPS by assessing apoptosis in cisp-treated macrophages and exploring the possibility of CTPS as a novel candidate for cancer treatment.

## 2. Results

### 2.1. Chemical Composition of CTPS

The monosaccharide composition and acidic polysaccharide content in CTPS are listed in Table 1. CTPS consists of glucose, glucosamine, fucose, and mannose. Notably, the glucose content is significantly higher than that of other sugars such as fucose, glucosamine, and mannose. The proportion of acidic polysaccharides in CTPS is 42.3%, which indicates a high content of acidic polysaccharides. To determine the characteristics of acidic polysaccharides in CTPS more accurately, FTIR spectroscopy was performed. The FTIR spectrum of CTPS indicated polysaccharide absorption peaks in the range of 4000–600 cm^−1^ (Appendix A). A wide absorption peak at 3410 cm^−1^ was designated as the O–H stretching vibration peak, and the relatively sharp peak at ~2943 cm^−1^ was attributed to the stretching vibration of C–H groups. A strong and sharp absorption peak at approximately 1068 cm^−1^ was assigned to C–C–C stretching vibration of carbonyl and neighboring carbons The absorption peaks at ~1705 and 1432 cm^−1^ highlighted the existence of the carbonyl groups, which indicated the characteristic IR absorption of uronic acids in CTPS. The presence of uronic acid is closely related to the antioxidant activity of the polysaccharide because of its chelating ability [24]. We therefore investigated the protective effects of CTPS on cisp-induced cytotoxicity.

### 2.2. Effects of CTPS on Cisp-Stimulated Lung Cancer Cells and Macrophage Cytotoxicity

To determine the maximum threshold concentration of CTPS, we first evaluated the cytotoxicity of CTPS in macrophage RAW264.7 cells. RAW264.7 cells were treated with serial doses of CTPS for 24 h. Figure 1A shows no effect of CTPS on cell viability. Next, we determined whether CTPS affected the viability of cisp-treated lung cancer cells (A549 and H460) and RAW264.7 cells. In A549 and H460 cells, cisp alone or in combination with CTPS induced significant cell death, whereas cytotoxicity was effectively suppressed by CTPS in cisp-treated RAW264.7 cells (Figure 1B). Based on the results, appropriate concentrations for cisp (15 µM) and CTPS (31.25 and 62.5 µg/mL) were determined and used in all subsequent experiments.

### 2.3. Inhibitory Effects of CTPS on Cisp-Stimulated Macrophage Apoptosis

To further confirm that CTPS protected macrophage viability in cisp-treated RAW264.7 cells, cytotoxicity was evaluated using annexin V/PI staining and a morphological image monitoring system. As shown in Figure 2, pretreatment with CTPS (31.25 and 62.5 µg/mL) led to a significant increase in cell viability when compared with the viability of the only cisp-treated group (Figure 2A). The only cisp-treated group was considered as the positive control. Concomitantly, in the cisp-treated group, the number of terminal deoxynucleotidyl transferase dUTP nick end labeling (TUNEL)-positive cells was higher than that in the control group, and this number was reduced via CTPS pretreatment (Figure 2B). Furthermore, cell penetration through the filter was effectively increased by CTPS (31.25 and 62.5 µg/mL) treatment compared with that in the control group (Figure 2C). Collectively, these results suggested that CTPS attenuated apoptosis and necrosis and mitigated morphological damage in cisp-treated RAW264.7 cells.

### 2.4. Effects of CTPS on the Mitochondrial Apoptotic Pathway in Cisp-Treated Macrophages

To elucidate the potential molecular pathways underlying CTPS-induced cytoprotective action, we examined the protein expression levels of pro-apoptotic (Bax) and anti-apoptotic (Bcl-2) proteins, cytochrome *c*, PARP, caspase-3, -8, and -9, which are all related to cell death, in cisp-treated RAW264.7 cells. As shown in Figure 3A,B, exposure to cisp resulted in an upregulation of Bax and cytosolic cytochrome *c* and downregulation of Bcl-2 in the cells. In contrast, pretreatment with CTPS (31.25 and 62.5 μg/mL) notably reduced the expression of Bax and cytosolic cytochrome *c* and increased the expression of Bcl-2 compared to cells treated with cisp alone. Next, we investigated the involvement of caspase-3, -8, and -9 in the cisp-induced cytotoxic effects observed in RAW264.7 cells. As shown in Figure 3C, CTPS treatment markedly inhibited the cisp-triggered activation of caspases-3, -8, and -9 in the cells. Furthermore, greater PARP cleavage was observed in the cisp-treated group than in the control group, whereas pretreatment with CTPS significantly suppressed the PARP cleavage in cisp-treated cells. These findings suggested that CTPS inhibited the mitochondrial apoptotic pathways in cisp-induced cytotoxicity, thereby promoting macrophage survival.

### 2.5. Effects of CTPS on ROS Production and MTP Loss in Cisp-Treated Macrophages

We next evaluated whether the CTPS (31.25 and 62.5 µg/mL)-induced cytoprotective action correlated with reactive oxygen species (ROS) generation and mitochondrial transmembrane potential (MTP) loss in macrophages. As shown in Figure 4A, cisp treatment resulted in an increase in ROS levels compared to that in the control group, whereas the increased ROS levels induced by cisp were significantly attenuated by CTPS pretreatment. In addition, a significant loss of MTP was observed in cisp-treated RAW264.7 cells (Figure 4B), which was attenuated by CTPS pretreatment. Subsequently, we found that the measured mitochondrial ROS level was significantly increased by cisp treatment, while CTPS pretreatment markedly inhibited this increase at dose of 62.5 µg/mL (Figure 4C). These findings suggested that the CTPS exerted cytoprotective effects via inhibiting the mitochondrial apoptotic pathways, reducing the ROS level and MTP loss induced by cisp treatment.

### 2.6. Effects of CTPS on Cisp-Stimulated BMDM Cytotoxicity

To confirm the cytoprotective effect of CTPS on primary macrophages, we pretreated bone marrow-derived macrophages (BMDMs) with CTPS before cisp treatment and examined the subsequent effects. In agreement with the results obtained using RAW264.7 cells, pretreatment with CTPS induced a significant increase in BMDM viability compared to BMDMs subjected to cisp treatment alone (Figure 5A). Additionally, CTPS treatment markedly inhibited the activation of cisp-triggered caspase-3, -8, and -9 in BMDMs, suggesting that CTPS attenuated the cell damage in cisp-treated macrophages (Figure 5B,C). These findings suggest that CTPS may act as a strong cytoprotective agent against cisp-induced cytotoxicity in immune cells.

### 2.7. Protective Effects of CTPS on Cisp-Induced Immunotoxicity in Mice with Lung Metastasis

To investigate whether CTPS had synergistic and protective effects against cisp, CTPS was orally administered to lung metastasis-induced mice in combination with cisp. After cisp injection on day 8, the body weight reduced markedly in the cisp-treated group; however, CTPS treatment alleviated the cisp-induced weight loss on day 17 (Figure 6A). Furthermore, the number of metastatic colonies in the lung tissue was significantly reduced in the CTPS and cisp co-treated group compared to that in the cisp-treated group (Figure 6B). We speculated that the anti-metastatic effects of CTPS may be related to its protective effect on immune cells against cisp-induced cytotoxicity. The flow cytometry data revealed that the absolute number of CD4^+^CD44^+^ and CD8^+^CD44^+^ cells increased in the CTPS and cisp co-treatment group than in the cisp-treated group (Figure 6C). These results indicated that co-treatment of CTPS and cisp may act synergistically to increase anticancer activity.

### 2.8. Effects of CTPS on Serum Biochemical Markers in Mice with Lung Metastasis

An increase in GOT, GPT, CRE, and BUN levels in serum is generally observed in cisp-treated mice, and could be considered as a hepatotoxicity and nephrotoxicity marker. Therefore, we investigated whether CTPS reduced the levels of these biochemical markers in cisp-treated mice. The serum levels of CRE and BUN, the early biomarkers of renal injury, were significantly increased in the cisp-treated group compared to the cancer only group. However, CTPS and cisp co-treatment decreased the serum levels of CRE and BUN (Figure 7). The serum levels of GOT and GPT, the biomarkers of hepatotoxicity, did not alter after cisp treatment in mice.

## 3. Discussion

Epidemiological evidence indicates that cancer has been the leading cause of mortality and morbidity for decades. Among the various treatment modalities, chemotherapy is considered the main treatment for cancer [25,26]. However, chemotherapeutic agents inevitably cause adverse effects such as non-specific cytotoxicity in normal organs and tissues in the body, which can lead to neurotoxicity, cardiotoxicity, hepatotoxicity, and hematological toxicity [15,27,28]. Among the chemotherapy agents, cisp—also known as *cis*-diamminedichloroplatinum (II)—is widely used for treating cancers of lungs, ovary, breasts, and brain [23,29]. However, because of its adverse effects and development of drug resistance, the efficacy of cisp is limited. Therefore, finding an effective approach to alleviate the side effects of chemotherapeutic drugs remains imperative [30].

Several recent studies on the use of natural products in preclinical and clinical trials have shown potential activity in reducing the adverse effects caused by chemotherapeutic drugs [16]. In particular, plant-derived acidic polysaccharides are considered as alternatives to chemical agents owing to their effective antioxidant activity [31,32]. Acidic polysaccharides generally contain considerable amount of uronic acid. Uronic acid possess effective chelating activity, and is therefore considered an important indicator of the antioxidant activity of polysaccharides [24]. In this study, we found that acidic polysaccharides are major components of CTPS, and FTIR data demonstrated that CTPS contains a considerable amount of uronic acid. Based on this, we hypothesized that CTPS could exert protective effects on immune cells against cisp-induced oxidative stress and cell death. Therefore, we performed in vitro and in vivo experiments toward three specific aims: (1) to determine whether CTPS exerts cytoprotective effects against cisp-induced cytotoxicity in macrophages (in vitro); (2) to confirm the molecular mechanisms underlying the cytoprotective effect of CTPS by inhibiting cisp-induced apoptotic pathways; (3) to investigate whether CTPS exerts protective effects against cisp-induced immunotoxicity and nephrotoxicity in lung metastasis-induced mice (in vivo).

First, we observed that CTPS and cisp co-treatment showed significant cytotoxicity in A549 and H460 lung cancer cells; however, in cisp-treated macrophages, the cytotoxicity was inhibited by CTPS co-treatment. Second, as demonstrated in recent studies showing the molecular mechanisms of the apoptotic death in cancer [33,34,35], the cytoprotective action of CTPS was mediated via suppressing the cisp-induced apoptotic pathways such as the upregulation of Bax, cytosolic cytochrome *c*, PARP cleavage, caspase-3, -8, and -9, and the downregulation of Bcl-2. These findings are consistent with those from earlier studies that reported the cytoprotective effects of polysaccharides derived from natural products in apoptotic cell death [36,37,38]. To further identify the mediators responsible for the cytoprotective action of CTPS on cisp-treated macrophages, we showed that CTPS suppressed the cisp-induced apoptotic pathways via reducing ROS production and MTP loss. These findings are similar to those of earlier studies, which reported the role of ROS, MTP, and DNA fragmentation in the initiation and execution of apoptosis via mitochondria [39,40,41]. In view of these findings, it may be speculated that the cytoprotective effect of CTPS against cisp-induced cytotoxicity is mediated by suppressing apoptotic pathways and regulating the mitochondrial dysfunction in cisp-treated macrophages [39,42]. To further validate the in vitro results, we examined the effects of CTPS against cisp-induced immunotoxicity and biochemical marker of hepato- and nephro-toxicity in lung metastasis-induced mice (in vivo). CTPS treatment significantly inhibited the metastatic colonies and bodyweight loss as well as immunotoxicity in splenic T cells compared to the control group of lung metastasis-induced mice. CTPS treatment also significantly decreased the cisp-induced increase in CRE and BUN serum levels, consistent with previous studies that demonstrated the role of natural products in reducing nephrotoxic side effects of chemotherapeutic drugs [43]. These data indicate that CTPS can attenuate cisp-induced immunotoxicity and nephrotoxicity. However, further confirmation of the cytoprotective effects of CTPS in human macrophages, such as the human peripheral blood monocyte-derived macrophages, is required for an in-depth understanding of these effects in immune cells. Furthermore, the cytoprotective effects of CTPS on nephrotoxicity, which is considered as the most common side effect of cisp treatment, should be investigated using in vitro and in vivo models. In addition, the antioxidant systems, including antioxidant enzymes (e.g., superoxide dismutase, catalase, and glutathione peroxidase) and their transcription factors (nuclear factor erythroid 2-related factor 2), are closely related to the cytoprotective activity of natural ingredients against chemotherapeutic agents [44,45]. In this regard, to utilize CTPS as a chemotherapeutic supplement, further studies on the role and the underlying mechanism of CTPS activity against other toxic chemotherapeutic agents are needed.

In conclusion, the present study provides strong evidence that CTPS exerts cytoprotective effects against cisp-induced apoptotic death in vitro and in vivo. CTPS reduced the levels of cisp-induced mitochondrial apoptotic mediators in macrophages. Furthermore, oral administration of CTPS attenuated the cisp-induced loss in bodyweight and immunotoxicity in splenic T cells and increased the serum CRE and BUN levels in mice. Therefore, CTPS can be a potential candidate for use as a therapeutic supplement to alleviate the adverse effects induced by chemotherapeutic drugs. However, to use CTPS as a chemotherapeutic supplement further studies regarding CTPS toxicity are required.

## 4. Materials and Methods

### 4.1. Preparation of CTPS

*Cudrania tricuspidata* fruits were obtained from the Geumsan herbal market in the Chungnam Province of Korea, freeze-dried (VD-800F; Taitec, Nishikata, Saitama, Japan), and mashed into small particles. The lyophilized granules (100 g/L) were extracted in deionized water (DW) at 90 °C for 2 h and vacuum filtered using Whatman filter paper 42. To precipitate the filtrate, 70% ethanol was added to the filtered solution and the solution was stirred at 4 °C for 12 h. The solution was centrifuged at 10,000× *g* for 15 min and dialyzed (molecular weight cutoff: 10–12 kDa) against DW to eliminate monosaccharides and other components. Following dialysis, the precipitate was lyophilized to obtain CTPS.

### 4.2. Analysis of Monosaccharide Composition and Acidic Polysaccharides of CTPS

Monosaccharides obtained by acid hydrolysis of CTPS were analyzed via high-performance anion-exchange chromatography (HPAEC). HPAEC was performed on a Dionex ICS-5000 system (Dionex Corp., Sunnyvale, CA, USA) coupled with a CarboPac PA1 column (guard column 4.5 × 250 mm). Monosaccharides were detected using a pulsed amperometric detector and were isocratically eluted using 16 mM NaOH. To measure the acidic polysaccharide content in CTPS, the method described by Bittere [46] was performed with modifications. The reaction mixture, including 0.5 mL of CTPS (1 mg/mL), 0.25 mL of 0.125% carbazole solution (dissolved in 0.1% ethanol), and 3 mL of sulfuric acid, was incubated for 30 min at 25 °C. Galacturonic acid was used as a standard and the absorbance of acidic polysaccharides was measured at 525 nm using a microplate reader (Zenyth 3100; Anthos Labtec Instruments GmbH, Salzburg, Austria). Carbazole, sulfuric acid, and galacturonic acid were obtained from Sigma-Aldrich (St. Louis, MO, USA).

### 4.3. Fourier Transform Infrared (FTIR) Spectroscopy

The FTIR spectrum of CTPS was assessed on a VERTEX 70v vacuum FTIR spectrometer (Bruker AXS, Billerica, MA, USA) with a range of 600–4000 cm^−1^. CTPS powder (1 mg) was mixed with 100 mg of KBr powder (Sigma-Aldrich) and then pressed into 1 mm pellets for further infrared spectrometric analysis.

### 4.4. Reagents and Antibodies

A fluorescein isothiocyanate (FITC)-annexin V/propidium iodide (PI) kit was purchased from BD Biosciences (San Diego, CA, USA). Cisp and 3-(4,5-dimethylthiazol-2-yl)-2,5-diphenyltetrazolium bromide (MTT) were procured from Sigma-Aldrich. Hybond-polyvinylidene difluoride (PVDF) membranes, goat anti-rabbit IgG horseradish peroxidase (HRP)-conjugated antibody, goat anti-mouse IgG HRP-conjugated antibody, and an electrochemiluminescence (ECL) solution were purchased from Merck (KGaA, Darmstadt, Germany). Protein extraction solution (RIPA Lysis and Extraction Buffers) was purchased from Thermo Fisher Scientific (Carlsbad, CA, USA). Antibodies specific to Bax, Bcl-2, cytochrome *c*, cleaved caspase-3, cleaved caspase-8, cleaved caspase-9, cleaved poly (ADP-ribose) polymerase (PARP), and β-actin were obtained from Cell Signaling Technology (Danvers, MN, USA). BV510-conjugated live/dead staining kit, V450-conjugated anti-cluster of differentiation 44 (CD44), FITC-conjugated anti-CD4, PE-conjugated anti-CD3, and PE-Cy7-conjugated anti-CD8 Abs were purchased from (BD Bioscience).

### 4.5. Cell Culture

Lung cancer cell lines A549 and H460 and murine macrophage cell line RAW264.7 were purchased from the Korean Cell Line Bank (Seoul, Korea). A549, H460, and RAW264.7 cells were cultured in Dulbecco’s modified Eagle’s medium (DMEM; Welgene, Gyeongsan, Korea) supplemented with 10% fetal bovine serum (Thermo Fisher Scientific), 100 U/mL penicillin, and 100 U/mL streptomycin at 37 °C with 5% CO_2_. A549 and H460 cells were used in passage 10–13, and RAW264.7 cells were used in passage 60. BMDMs were isolated from the femur and tibia bones of 6–8-week-old C57BL/6 mice and differentiated after four days in a medium containing macrophage colony-stimulating factor (25 ng/mL; R&D Systems, Minneapolis, MN, USA) at 37 °C with 5% CO_2_.

### 4.6. Cytotoxicity Analysis

Cytotoxicity was assessed by performing the MTT (Sigma-Aldrich) assay and using the annexin V/PI staining kit (BD Biosciences) according to the manufacturer’s instructions. To perform the MTT assay, cells were seeded in 96-well plates at 2 × 10^4^ cells/well. MTT solution (5 mg/mL in distilled water) was prepared immediately prior to use and filtered through a 0.22 µm syringe filter (Merck Millipore, Billerica, MA, USA). The cells were treated with CTPS in the absence and presence of cisp. CTPS were pretreated 2 h prior to cisplatin treatment. After 24 h, the culture medium in each well was replaced with 80 µL of complete DMEM, to which 20 µL of the MTT stock solution was added (final MTT concentration: 1 mg/mL). The cells were incubated for 1 h at 37 °C, the medium was removed, and the cells were washed with phosphate-buffered saline (PBS). The formazan crystals were dissolved by adding 100 µL dimethyl sulfoxide to each well, and the absorbance was measured at 570 nm using a microplate reader (Zenyth 3100). To perform Annexin V/PI staining, the cells were seeded in 12-well plates at 1 × 10^5^ cells/well and treated with CTPS in the absence and presence of cisp. After 24 h, the harvested cells were washed with PBS and stained with FITC-conjugated with Annexin V and PI. Stained cells were analyzed using the FACSVerse^TM^ flow cytometer (BD Biosciences) and the FlowJo software (TreeStar, Ashland, OR, USA).

### 4.7. Cell Migration Assay

Cells were seeded in a 12-well plate and incubated overnight at 37 °C with 5% CO_2_. The medium was removed from the plate, and the well surfaces were gently scraped using a sterilized wooden applicator stick (1 mm outer diameter; Baxter healthcare, Deerfield, IL, USA). The wells were gently washed with PBS and filled with complete DMEM. The cells were then treated with CTPS in the absence and presence of cisp and incubated for 24 h. Cell migration behavior was determined by taking snapshots with an inverted microscope (CKX53, Olympus, Tokyo, Japan) and using the VisionMate ST software (Thermo Fisher Scientific).

### 4.8. Immunoblotting

RAW264.7 cells were detached, centrifuged, and lysed using a RIPA Lysis and Extraction Buffer containing 1 mM phenylmethanesulfonyl fluoride and protease inhibitor cocktail. Protein concentrations were determined using the Bradford assay. Ten micrograms of protein was resolved via sodium dodecyl sulfate-polyacrylamide gel electrophoresis and then transferred onto a PVDF membrane. The membrane was blocked with 5% skim milk, incubated with the relevant Ab for 2 h, followed by incubation with an HRP-conjugated secondary Ab for 1 h at room temperature. The bands of target proteins (Bax, Bcl-2, cytochrome *c*, cleaved caspase-3, 8, 9, and PARP) were visualized using the ECL solution kit.

### 4.9. Measurement of ROS Levels

Intracellular ROS levels were measured by staining the cells with 2′,7′-dichlorodihydrofluorescein diacetate (H_2_DCFDA, Thermo Fisher Scientific). The cells were treated with 10 µM H_2_DCFDA for 30 min at 37 °C in the dark and then washed with PBS. Mitochondrial ROS levels were measured using MitoSOX mitochondrial superoxide indicator (Thermo Fisher Scientific). The cells were stained with 5 µM MitoSOX reagent for 10 min at 37 °C in the dark and then washed with PBS. The samples were immediately analyzed using the FACSVerse^TM^ flow cytometer, and data were processed using FlowJo software.

### 4.10. Assessment of MTP

MTP was assessed by measuring the retention of the lipophilic cationic dye 3,3′-dihexyloxacarbocyanine (DiOC_6_, Thermo Fisher Scientific) in mitochondria. Cells were harvested and incubated in a DiOC_6_ solution (10 nM in fresh medium) for 30 min at 37 °C in the dark. Next, the cells were washed and resuspended in PBS. Immediately after PBS washing, the MTP was measured by sorting the cells using the FACSVerse^TM^ flow cytometer, and the data were processed using FlowJo software.

### 4.11. TUNEL Staining

The TUNEL assay was performed using a kit containing deoxynucleotide fluorescein-12-dUTP (Takara, Kyoto, Japan). DNA labeled with fluorescein was measured using the FACSVerse^TM^ flow cytometer, and the data were analyzed using FlowJo software (version 10).

### 4.12. Animals

Specific pathogen-free, 7-week-old female C57Bl/6NCrljOri strain C57BL/6 mice weighing 16–18 g were purchased from Orient Bio Inc., Seoul, Korea. The mice were acclimatized to the following controlled conditions: temperature (23 ± 2 °C), humidity (55 ± 5%), and light (12 h light/dark cycle). They were divided into groups such that the average weight of mice in each group was statistically identical. All experimental procedures were approved by the Institutional Animal Care and Use Committee of Korea Atomic Energy Research Institute (KAERI-IACUC-2020-002).

### 4.13. Development of Lung Metastasis via B16BL6 Treatment

The mice were divided into four groups (*n* = 6 per group): (1) normal, (2) cancer only, (3) cancer + cisp, and (4) cancer + cisp + CTPS. The mice in groups 2–4 were intravenously injected with 3 × 10^5^ B16BL6 cells in 200 µL PBS through the tail vein on day 1. CTPS (200 mg/kg) was administered orally every day after B16BL6 inoculation until the end of the experiments. Cisp (10 mg/kg) was injected intraperitoneally on day 8. The mice were euthanized 17 days post tumor cell inoculation. The tumor colonies in the lung tissue were counted under a dissection microscope.

### 4.14. Analysis of Immunotoxicity in Splenic T Cells

Whole spleen tissues were harvested from the mice of each group. Single-cell splenocyte suspensions were prepared using a conventional method [47]. The cells were pre-incubated with 0.5% bovine serum albumin in PBS for 30 min and stained with BV510-conjugated live/dead staining kit, V450-conjugated anti-CD44, FITC-conjugated anti-CD4, PE-conjugated anti-CD3, and PE-Cy7-conjugated anti-CD8 Abs for 30 min at 4 °C. After washing, the cells were counted via flow cytometry (MACSQuant VYB, Miltenyi Biotec, Bergish Galdbach, Germany) and analyzed using FlowJo software.

### 4.15. Analysis of Biochemical Marker in Serum

Blood samples were collected via cardiac puncture. Serum was separated by centrifugation at 12,000 rpm for 15 min. Changes in the activities of glutamate oxaloacetate transaminase (GOT), glutamate pyruvate transaminase (GPT), blood urea nitrogen (BUN), and creatine (CRE) were measured in the serum using an automatic biochemical analyzer (DRI-CHEM 4000i, Fujifilm, Tokyo, Japan).

### 4.16. Statistical Analysis

In vitro experiments were performed at least three times in triplicate. In vivo experiments were performed two times. The levels of significance for comparison between samples were determined using one-way ANOVA followed by Tukey’s multiple comparison test using GraphPad Prism 8 (GraphPad Software, San Diego, CA, USA). The data are expressed as the mean ± standard deviation. Results with * *p* < 0.05, ** *p* < 0.01, and *** *p* < 0.001 were considered significant.

## Figures and Tables

**Figure 1 ijms-22-07512-f001:**
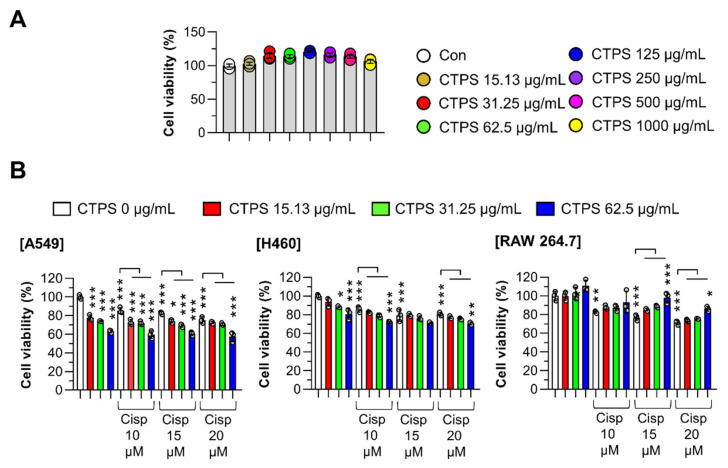
Effects of CTPS on the proliferation and viability of cisp-treated macrophages. (**A**) Cell viability after CTPS treatment with various doses (0, 15.625, 31.25, 62.5, 125, 250, 500, and 1000 µg/mL) for 24 h in RAW264.7 cells. (**B**) After CTPS pretreatment for 2 h with various doses (15.625, 31.25, and 62.5 µg/mL), the cell proliferation was analyzed in the presence of cisp (0, 10, 15, and 20 µM) for 24 h. The data are expressed as the mean ± standard deviation (*n* = 3). * *p* < 0.05, ** *p* < 0.01 or *** *p* < 0.001. CON: untreated cells; CTPS, *Cudrania tricuspidata*-derived polysaccharides; MTT, 3-(4,5-dimethylthiazol-2-yl)-2,5-diphenyltetrazolium bromide; cisp, cisplatin.

**Figure 2 ijms-22-07512-f002:**
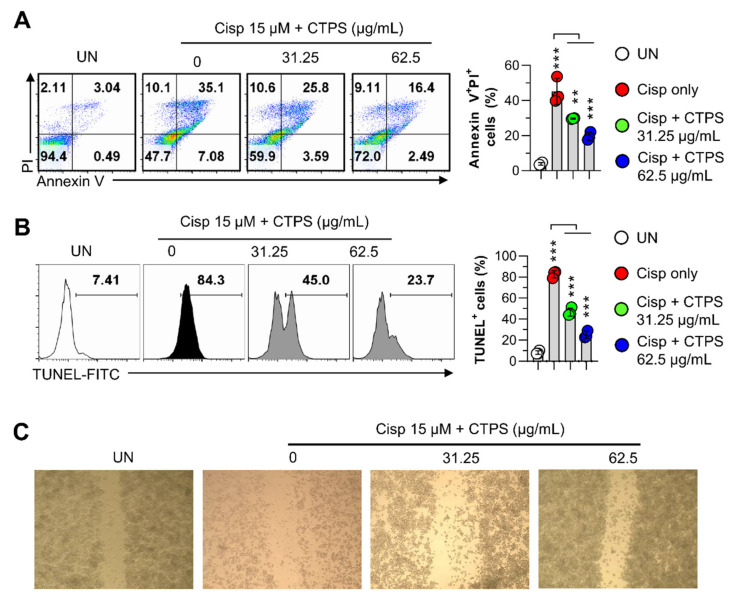
CTPS treatment decreases the cisp-induced macrophage apoptosis. (**A**) RAW264.7 cells were pretreated with CTPS (31.25 and 62.5 µg/mL) for 2 h and then treated with 15 µM cisplatin for 24 h. This was followed by flow cytometry analysis after Annexin V/PI (A) or TUNEL staining (**B**). The cells were analyzed via flow cytometry using the FACSVerse^TM^ flow cytometer. (**C**) Cell migration was observed under a microscope. The data are expressed as the mean ± standard deviation (*n* = 3). ** *p* < 0.01 or *** *p* < 0.001. UN, untreated cells; CTPS, *Cudrania tricuspidata*-derived polysaccharides; cisp, cisplatin; PI, propidium iodide.

**Figure 3 ijms-22-07512-f003:**
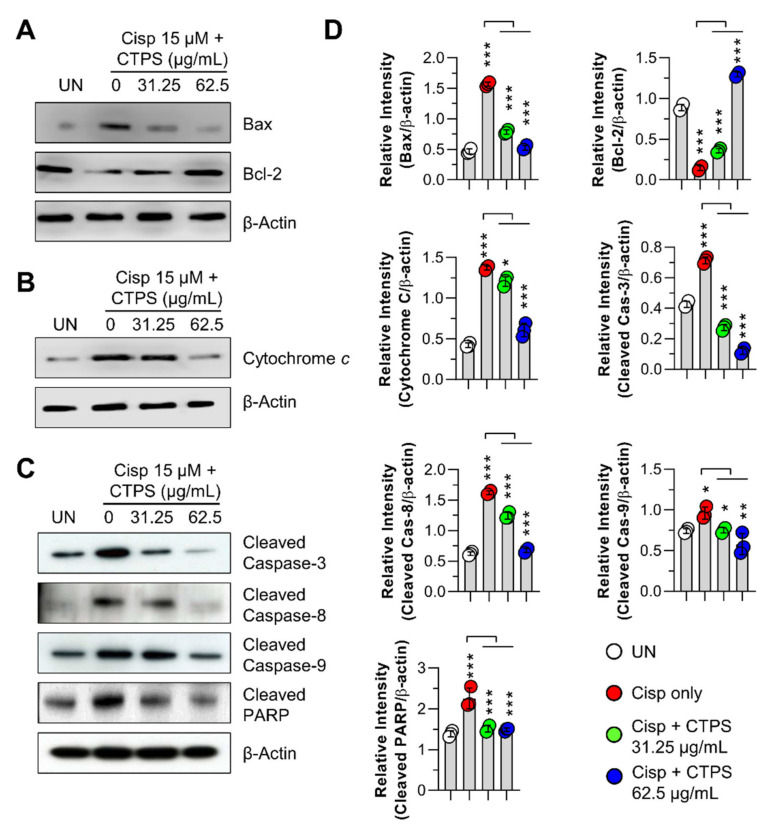
CTPS protects macrophages against cisplatin-induced cytotoxicity. Following pretreatment with CTPS (31.25 and 62.5 µg/mL) for 2 h, non- or CTPS-treated RAW264.7 cells were stimulated with 15 µM cisplatin for 24 h and then lysed. Proteins in the cell lysate were evaluated via western blotting. (**A**) Protein expression of Bax and Bcl-2 in cytosolic fractions. (**B**) Protein expression of cytochrome *c* in the cell lysate. (**C**) Protein expression of cleaved caspase-3, 8, 9, and PARP in cell lysate. β-actin was used as the loading control for cytosolic fractions. (**D**) Western blotting data are representative of three independent experiments. The data were analyzed using ImageJ program to compare the relative band intensity of each protein. The data are shown as the mean ± standard deviation (*n* = 3). * *p* < 0.05, ** *p* < 0.01, or *** *p* < 0.001. UN, untreated cells; CTPS, *Cudrania tricuspidata*-derived polysaccharides; cisp, cisplatin.

**Figure 4 ijms-22-07512-f004:**
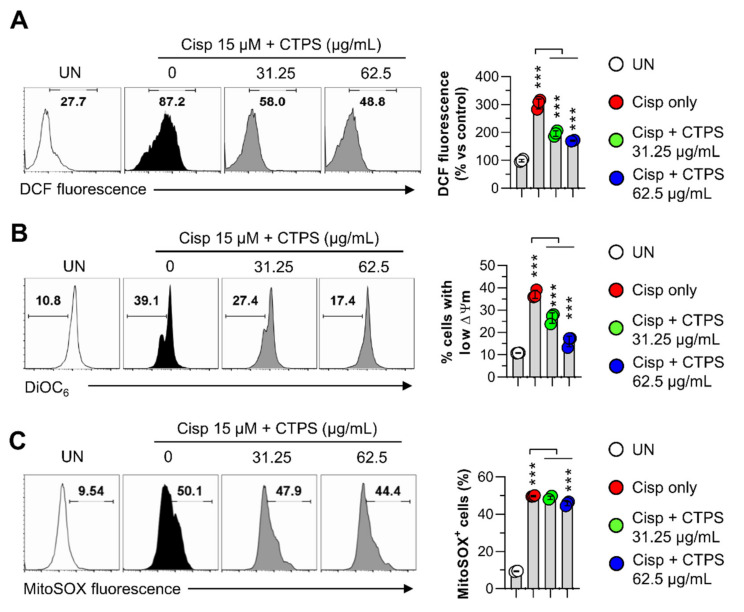
CTPS reduces the ROS production and loss of mitochondrial transmembrane potential in cisplatin-treated macrophages. Following pretreatment with CTPS (31.25 and 62.5 µg/mL) for 2 h, non- or CTPS-treated RAW264.7 cells were treated with 15 µM cisplatin for 24 h followed by H_2_DCFDA measurement (**A**), DiOC_6_ staining (**B**), or MitoSOX staining (**C**). The cells were analyzed via flow cytometry using FACSVerse^TM^ flow cytometer. The histograms are representative data of three independent experiments. The data are expressed as the mean ± standard deviation (*n* = 3). *** *p* < 0.001. UN: untreated cells; CTPS, *Cudrania tricuspidata*-derived polysaccharides; cisp, cisplatin; H_2_DCFDA, 2′,7′-dichlorodihydrofluorescein diacetate; DiOC_6_, 3,3′-dihexyloxacarbocyanine iodide; TUNEL, terminal deoxynucleotidyl transferase dUTP nick end labeling; FITC, fluorescein isothiocyanate.

**Figure 5 ijms-22-07512-f005:**
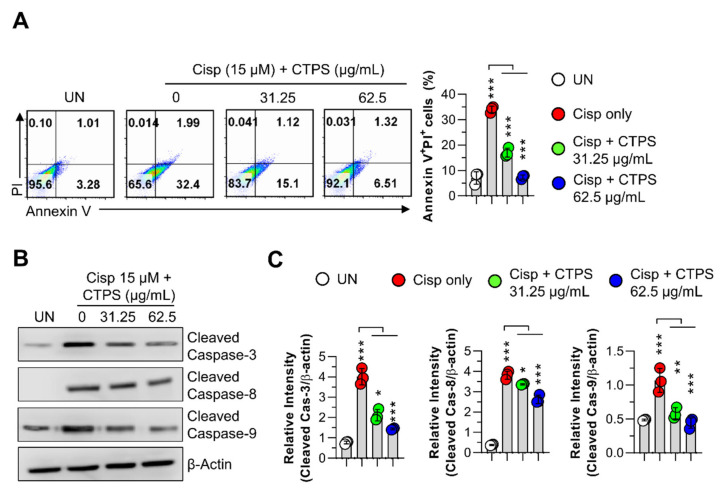
CTPS decreases cisplatin-induced apoptosis in BMDM. (**A**) BMDMs were pretreated with CTPS (31.25 and 62.5 µg/mL) for 2 h and then treated with 15 µM cisplatin for 24 h and analyzed via flow cytometry after performing Annexin V/PI staining. Cells were analyzed via flow cytometry using the FACSVerse^TM^ flow cytometer. The dot plots are representative data of three independent experiments. (**B**) Following pretreatment with CTPS (31.25 and 62.5 µg/mL) for 2 h, non- or CTPS-treated BMDMs were stimulated with 15 µM cisplatin for 24 h and lysed. Proteins in cell lysate were evaluated via western blotting. Protein expression of cleaved caspase-3, 8, and 9 in the cell lysate. β-actin was used as the loading control for cytosolic fractions. (**C**) The relative band intensity of each protein is expressed as a percentage. The data are expressed as the mean ± standard deviation (*n* = 3). * *p* < 0.05, ** *p* < 0.01, or *** *p* < 0.001. UN: untreated cells; CTPS, *Cudrania tricuspidata*-derived polysaccharides; cisp, cisplatin; BMDMs, bone marrow-derived macrophages; PI, propidium iodide.

**Figure 6 ijms-22-07512-f006:**
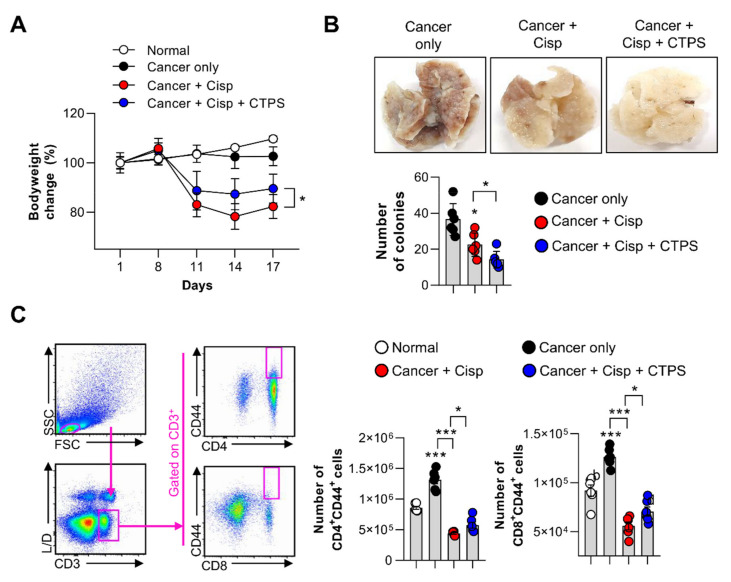
CTPS and cisplatin increase the anti-metastasis activity in C57BL/6 mice. (**A**) Change in body weight was monitored after the intravenous injection of mice with B16BL6 cells. CTPS (200 mg/kg) was orally administered daily. Cisplatin (10 mg/kg) was intraperitoneally administered on day 8. (**B**) Photographic representative images of lung tissues from each group of mice. The lung tissues were removed 17 days after injection. The metastatic colonies in the lung tissue were counted using a dissection microscope. (**C**) Single cells were isolated from the spleen of mice from each group and stained with anti-CD3, anti-CD4, anti-CD8, anti-CD44 monoclonal antibodies and subjected to the live/dead cell staining followed by flow cytometry analysis. The gating strategy is shown in the left panel. Bar graph shows the absolute number of CD4^+^CD44^+^ or CD8^+^CD44^+^ T cells. Data are expressed as the mean ± standard deviation (*n* = 6). * *p* < 0.05 and *** *p* < 0.001 represent significant differences. CTPS, *Cudrania tricuspidata*-derived polysaccharides; cisp, cisplatin; CD3, cluster of differentiation; FSC, forward scatter; SSC, side scatter.

**Figure 7 ijms-22-07512-f007:**
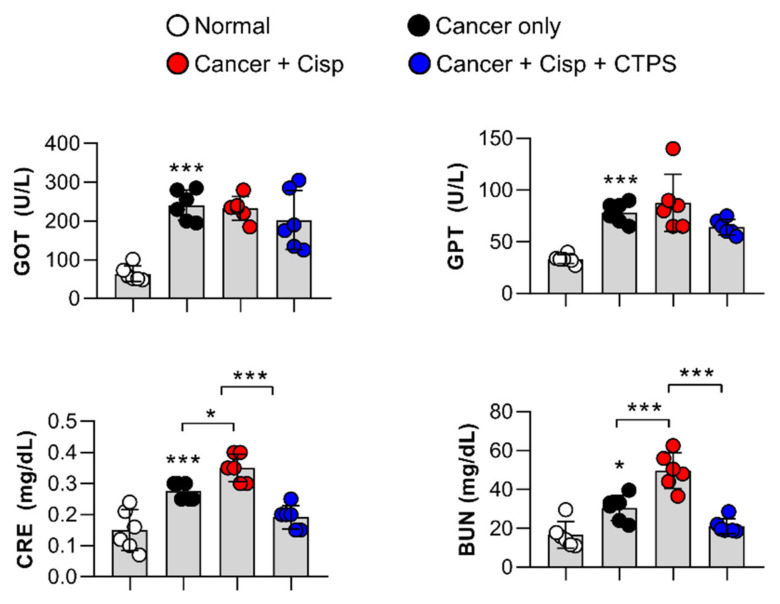
CTPS reduced the serum creatine and BUN levels in cisplatin-treated mice. Serum levels of GOT, GPT, CRE, and BUN were measured using an automatic biochemical analyzer. Data are expressed as the mean ± standard deviation (*n* = 6). * *p* < 0.05 and *** *p* < 0.001 represent significant differences. CTPS, *Cudrania tricuspidata*-derived polysaccharides; cisp, cisplatin; GOT, glutamate oxaloacetate transaminase; GPT, glutamate pyruvate transaminase; BUN, blood urea nitrogen; CRE, creatine.

**Table 1 ijms-22-07512-t001:** Content of monosaccharides and acidic polysaccharides of CTPS.

Group	Contents (g/100 g)
Fucose	0.004 ± 0.0048
Glucosamine	0.081 ± 0.0015
Glucose	17.454 ± 0.1550
Mannose	0.240 ± 0.0005
Acidic polysaccharides	42.325 ± 0.7889

The data are expressed as the mean ± standard deviation (*n* = 3).

## Data Availability

Not applicable.

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
