# Peer review of "Protective Effect of Polysaccharides Extracted from Cudrania tricuspidata Fruit against Cisplatin-Induced Cytotoxicity in Macrophages and a Mouse Model"

_ijms, 2021, doi:10.3390/ijms22147512_

Round 1

Reviewer 1 Report

Protective effect of polysaccharides extracted from Cudrania tricuspidata fruit against cisplatin-induced cytotoxicity in vitro and in vivo

In the present study, the authors conducted this study to explore the cytoprotective effect of Cudrania tricuspidata fruit-derived polysaccharides (CTPS) against cisplatin-induced cytotoxicity in vitro and in vivo models. In my opinion, the study is interesting and innovative. However, I have some comments:

Comment (1): Some parts of the text are confusing and needs a thorough revision of English.

Comment (2): Abstract. The background topic is not present and there is not a clear statement about the aim of this study. I recommend to including a sentence about the main objectives of the study.

Comment (3): Introduction. There is a brief review of existing knowledge and relevance of study. However, I would like to have some information about the side effects of using cisplatin to understanding the aim of the work.

Comment (4): Results. A lot of information are included but there are some comments.

- Page 3: Change the following sentence “…CTPS treatment of RAW264.7 cells did not affect… to “RAW264.7 cells were treated with serial doses of CTPS for XX h. Figure 1A shows no effect of CTPS on cell viability.”

Comment (5): Discussion. Great information in this section but need a proofreading to improve it well. The authors concluded their works at the end systematically.

Comment (6): Materials and Methods.

- Could you please write the city, state, country for each company from where you got your materials?

Comment (7): Figures and legends. Legends are adequate and figures are necessary to understand the results obtained.

- I recommend to the authors to re-present Figure 1B again because it is hard to see the differences of the values. It is better to change the bar colors.

Author Response

[Reviewer 1]

In the present study, the authors conducted this study to explore the cytoprotective effect of Cudrania tricuspidata fruit-derived polysaccharides (CTPS) against cisplatin-induced cytotoxicity in vitro and in vivo models. In my opinion, the study is interesting and innovative. However, I have some comments:

  • Answer: The authors would like to express a sincere appreciation to your kind review. We will try our best to make adequate correction based on your comments.

Comment (1): Some parts of the text are confusing and needs a thorough revision of English.

  • Answer: Thank you very much for your important comments. We proofread our manuscript following professional help and enclosed a certificate of English editing. The revised sentences and typo are marked with red.

Comment (2): Abstract. The background topic is not present and there is not a clear statement about the aim of this study. I recommend to including a sentence about the main objectives of the study.

  • Answer: As you pointed out, we added the background topic in abstract. Additionally, we revised the sentence about main objective of this study for clear legibility of readers. The revised parts were marked with red (Line 15-17).

Comment (3): Introduction. There is a brief review of existing knowledge and relevance of study. However, I would like to have some information about the side effects of using cisplatin to understanding the aim of the work.

  • Answer: The reasonable points that you have suggested us to review were truly helpful to improving the quality of this study. We added the information regarding side effect of cisplatin in introduction section (Line 47-51).

Comment (4): Results. A lot of information are included but there are some comments. Page 3: Change the following sentence “…CTPS treatment of RAW264.7 cells did not affect… to “RAW264.7 cells were treated with serial doses of CTPS for XX h. Figure 1A shows no effect of CTPS on cell viability.”

  • Answer: Thank you for your kind comments. As you pointed out, we revised the sentence from “CTPS treatment of RAW264.7 did not affect cell viability when tested over a range of doses (Figure 1A)” to “RAW264.7 cells were treated with serial doses of CTPS for 24 h. Figure 1A shows no effect of CTPS on cell viability”.

Comment (5): Discussion. Great information in this section but need a proofreading to improve it well. The authors concluded their works at the end systematically.

  • Answer: As above mentioned, we proofread our manuscript and attached certificate of English editing.

Comment (6): Materials and Methods. Could you please write the city, state, country for each company from where you got your materials?

  • As you suggested, we checked the company information of all materials. We marked the city and state information in case of company located in USA, while only city and country information were marked in case of company located in other country. All revised parts are marked with red.

Comment (7): Figures and legends. Legends are adequate and figures are necessary to understand the results obtained. I recommend to the authors to re-present Figure 1B again because it is hard to see the differences of the values. It is better to change the bar colors.

  • Following your suggestion, we changed the bar graph color of each group in Figure 1B.

Reviewer 2 Report

This is an interesting report showing the cytoprotective effect of CTPS in normal and healthy cells after treatment with cisplatin. Authors also addressed the investigation of the molecular pathways involved in the cytoprotection activity and underlined the molecular mechanisms that are potentially responsible for this effect. Moreover, an in vivo study further corroborated the author's hypothesis. 

As such, CTPS emerges as a possible therapeutic supplement for cancer therapy, specifically when cisplatin is used. 

The work is organized in a clear manner and the results are presented clearly.

However, this referee raises the following criticisms:

-English shall be revised to fix a number of typos and sentences that are difficult to read. 

-A larger panel of human healthy (or normal) cells shall be used to sustain the main hypothesis. Indeed, authors only tested one macrophage line, i.e. RAW264.7, and BMDM. Other cells shall be tested, and/or other toxic chemotherapeutics with a different mechanism of action than cisplatin shall be monitored. In the present form, the results are a bit preliminary.

-The title should be focused on the target cell lines, as in the present form it is not very informative. 

Author Response

[Reviewer 2]

This is an interesting report showing the cytoprotective effect of CTPS in normal and healthy cells after treatment with cisplatin. Authors also addressed the investigation of the molecular pathways involved in the cytoprotection activity and underlined the molecular mechanisms that are potentially responsible for this effect. Moreover, an in vivo study further corroborated the author's hypothesis.

As such, CTPS emerges as a possible therapeutic supplement for cancer therapy, specifically when cisplatin is used. The work is organized in a clear manner and the results are presented clearly. However, this referee raises the following criticisms:

  • Answer: We really appreciate your review of our manuscript, and dedicated our best effort to revising the manuscript depending on your comments.

Q1: English shall be revised to fix a number of typos and sentences that are difficult to read.

  • Answer: The reasonable points that you have requested us were truly essential to improving the quality of this manuscript. We proofread our manuscript following professional help and attached a certificate of English editing. All revised parts are marked with red.

Q2: A larger panel of human healthy (or normal) cells shall be used to sustain the main hypothesis. Indeed, authors only tested one macrophage line, i.e. RAW264.7, and BMDM. Other cells shall be tested, and/or other toxic chemotherapeutics with a different mechanism of action than cisplatin shall be monitored. In the present form, the results are a bit preliminary.

  • Answer: The authors totally agree with your comments. We added the limitation of the present study that need to confirm the protective effects of CTPS on human derived macrophage in the end of discussion part. We also described the need for further study of possibility on protective effects of against other toxic chemotherapeutic agents in discussion part. We will focus on whether CTPS affects antioxidant enzymes and their transcription factor (Nrf2) as a different mechanism on our further study.
  • We also suggested the recommended further study regarding the nephrotoxicity in discussion part. Because nephrotoxicity is a common side effect of cisplatin-treated patients, which is related with both mitochondrial and non-mitochondrial pathways of apoptosis.
  • The added sentences (Line 295-306): However, further confirmation of the cytoprotective effects of CTPS in human macrophages, such as the human peripheral blood monocyte-derived macrophages, is required for an in-depth understanding of these effects in immune cells. Furthermore, the cytoprotective effects of CTPS on nephrotoxicity, which is considered as the most common side effect of cisp treatment, should be investigated using in vitro and in vivo models. In addition, the antioxidant systems, including antioxidant enzymes (e.g., superoxide dismutase, catalase, and glutathione peroxidase) and their transcription factors (nuclear factor erythroid 2-related factor 2), are closely related to the cytoprotective activity of natural ingredients against chemotherapeutic agents [44,45]. In this regard, to utilize CTPS as a chemotherapeutic supplement, further studies on the role and the underlying mechanism of CTPS activity against other toxic chemotherapeutic agents are needed.

Q3: The title should be focused on the target cell lines, as in the present form it is not very informative.

  • We would like to thank your helpful comments. We changed the title from “Protective effect of polysaccharides extracted from Cudrania tricuspidata fruit against cisplatin-induced cytotoxicity in vitro and in vivo” to “Protective effect of polysaccharides extracted from Cudrania tricuspidata fruit against cisplatin-induced cytotoxicity in macrophage and mouse model”

Reviewer 3 Report

I read with interest the manuscript entitled ijms-1273670 entitled “Protective effect of polysaccharides extracted from Cudrania tricuspidata fruit against cisplatin-induced cytotoxicity in vitro and in vivo” that has been submitted to IJMS Journal.

The authors suggest that CTPS induced cytoprotective action in alleviating the side effects induced by the chemotherapeutic drugs, by means of in vitro and in vivo experiments.

In my opinion, this paper is acceptable in IJMS journal with a major issue, as it follows:

  • The Authors performed measurement of Intracellular reactive oxygen species (ROS) levels by staining the cells with 2′,7′-dichlorodihydrofluorescein diacetate (H2DCFDA)

The Authors assessed mitochondrial transmembrane potential (MTP) by measuring the retention of the lipophilic cationic dye 3,3'- dihexyloxacarbocyanine (DiOC6)

  • Based on these results the Authors concluded that suggested that the CTPS exerted cytoprotective effects via inhibiting the mitochondrial apoptotic pathways, reducing the ROS level and MTP loss induced by cisplatin treatment

Considering that CTPS treatments reduced the loss of MTP caused by cisplatin, an assessment of the level of mitochondrial ROS should be performed, to better analyze the cytoprotective effect of CTPS on cisplatin - treated cells. In fact, it is known that not only intracellular ROS but also mitochondrial ROS induced by oxidative stress, can cause cell death by activating apoptotic pathways.

Author Response

[Reviewer 3]

I read with interest the manuscript entitled ijms-1273670 entitled “Protective effect of polysaccharides extracted from Cudrania tricuspidata fruit against cisplatin-induced cytotoxicity in vitro and in vivo” that has been submitted to IJMS Journal.

The authors suggest that CTPS induced cytoprotective action in alleviating the side effects induced by the chemotherapeutic drugs, by means of in vitro and in vivo experiments.

In my opinion, this paper is acceptable in IJMS journal with a major issue, as it follows:

Q1: The Authors performed measurement of Intracellular reactive oxygen species (ROS) levels by staining the cells with 2′,7′-dichlorodihydrofluorescein diacetate (H2DCFDA)

The Authors assessed mitochondrial transmembrane potential (MTP) by measuring the retention of the lipophilic cationic dye 3,3'- dihexyloxacarbocyanine (DiOC6)

Based on these results the Authors concluded that suggested that the CTPS exerted cytoprotective effects via inhibiting the mitochondrial apoptotic pathways, reducing the ROS level and MTP loss induced by cisplatin treatment

Considering that CTPS treatments reduced the loss of MTP caused by cisplatin, an assessment of the level of mitochondrial ROS should be performed, to better analyze the cytoprotective effect of CTPS on cisplatin - treated cells. In fact, it is known that not only intracellular ROS but also mitochondrial ROS induced by oxidative stress, can cause cell death by activating apoptotic pathways.

  • Answer: The authors really appreciate your clear summary of our study and helpful comments. Following your suggestion, we analyzed mitochondrial ROS level using MitoSOX mitochondrial superoxide indicator. We added these data in Figure 4C. The added methods and results are marked with red (Line 171-173, 418-421).

Round 2

Reviewer 1 Report

The Authors addressed all my previous comments in this improved version. I recommend to accept this MS in the present form.

Reviewer 2 Report

I am happy to see that the manuscript has improved also thanks to my suggestions. English style is improved as well, which render the manuscript fluent to be read and understood.

Reviewer 3 Report

For Authors:

In my opinion this paper is now acceptable in IJMS.

All suggestions have been fulfilled.

Best regards